# Sex differences in multimorbidity and polypharmacy trends: A repeated cross-sectional study of older adults in Ontario, Canada

**Colleen J. Maxwell**[1,2,3]*, **Luke Mondor**[2,3], **Anna J. Pefoyo Koné**[3,4], **David B. Hogan**[5], **Walter P. Wodchis**[2,3,6,7]

**1** Schools of Pharmacy and Public Health & Health Systems, University of Waterloo, Waterloo, Ontario, Canada, **2** ICES, Toronto, Ontario, Canada, **3** Health System Performance Network (HSPN), Toronto, Ontario, Canada, **4** Department of Health Sciences, Lakehead University, Thunder Bay, Ontario, Canada, **5** Division of Geriatric Medicine, Department of Medicine, Cumming School of Medicine, University of Calgary, Calgary, Alberta, Canada, **6** Dalla Lana School of Public Health, University of Toronto, Toronto, Ontario, Canada, **7** Institute for Better Health, Trillium Health Partners, Toronto, Ontario, Canada

* colleen.maxwell@uwaterloo.ca

**Data Availability Statement:** The dataset from this study is held securely in coded form at ICES. While legal data sharing agreements between ICES and data providers (e.g., healthcare organizations and

## Abstract

### Background

Multimorbidity is increasing among older adults, but the impact of these recent trends on the extent and complexity of polypharmacy and possible variation by sex remains unknown. We examined sex differences in multimorbidity, polypharmacy (5+ medications) and hyper-polypharmacy (10+ medications) in 2003 vs 2016, and the interactive associations between age, multimorbidity level, and time on polypharmacy measures.

### Methods and findings

We employed a repeated cross-sectional study design with linked health administrative databases for all persons aged ≥66 years eligible for health insurance in Ontario, Canada at the two index dates. Descriptive analyses and multivariable logistic regression models were conducted; models included interaction terms between age, multimorbidity level, and time period to estimate polypharmacy and hyper-polypharmacy probabilities, risk differences and risk ratios for 2016 vs 2003. Multimorbidity, polypharmacy and hyper-polypharmacy increased significantly over the 13 years. At both index dates prevalence estimates for all three were higher in women, but a greater absolute increase in polypharmacy over time was observed in men (6.6% [from 55.7% to 62.3%] vs 0.9% [64.2%-65.1%] for women) though absolute increases in multimorbidity were similar for men and women (6.9% [72.5%-79.4%] vs 6.2% [75.9%-82.1%], respectively). Model findings showed that polypharmacy decreased over time among women aged < 90 years (especially for younger ages and those with fewer conditions), whereas it increased among men at all ages and multimorbidity levels (with larger absolute increases typically at older ages and among those with 4 or fewer conditions).

government) prohibit ICES from making the dataset publicly available, access may be granted to those who meet pre-specified criteria for confidential access, available at www.ices.on.ca/DAS (email: das@ices.on.ca). The data used in this study are accessible to DAS clients (see https://www.ices.on.ca/DAS/Data). The authors of this study are ICES scientists, collaborating researchers or staff and their experience and/or expertise with the ICES databases and processes would be likely to facilitate data access and/or analyses relative to researchers less familiar with these databases or processes. Otherwise, the authors did not receive any special privileges in accessing the data from ICES relative to other researchers.

**Funding:** This research was supported by a grant from the Ontario Ministry of Health (MOH) to the Health System Performance Network (HSPN) and by ICES, which is funded by an annual grant from the Ontario Ministry of Health (MOH) and the Ministry of Long-Term Care (MLTC). Parts of this material are based on data and information compiled and provided by the Canadian Institute for Health Information (CIHI). The analyses, conclusions, opinions and statements expressed herein are solely those of the authors and do not reflect those of the funding or data sources; no endorsement is intended or should be inferred. The funders had no role in study design, data collection and analyses, decision to publish, or preparation of the manuscript.

**Competing interests:** The authors have declared that no competing interests exist.

## Conclusions

There are sex and age differences in the impact of increasing chronic disease burden on changes in measures of multiple medication use among older adults. Though the drivers and health consequences of these trends warrant further investigation, the findings support the heterogeneity and complexity in the evolving association between multimorbidity and polypharmacy measures in older populations.

## Introduction

The prevalence of multimorbidity (i.e., co-occurrence of 2+ chronic conditions) is projected to increase and emerge at progressively younger ages over the coming decades [1,2]. This is due in part to population aging, improved survival among adults with chronic conditions, and upward trends in select negative health behaviours (e.g., obesity, sedentary lifestyle) earlier in life [2–4]. Though multimorbidity estimates are increasing among most age groups [2,4–6], overall prevalence is highest (upwards of 75%) among adults aged 65 and older [2,6–9].

The potential for significant expansion in years lived with multimorbidity [1] has important implications for healthcare outcomes and costs [5,10]. As both the number and heterogeneity of co-occurring chronic conditions increases with age [6,7], older adults are more likely to be exposed to complex drug regimens that place them at risk for inappropriate use [11,12], treatment burden [13], nonadherence [14] and adverse drug events [15–18]. As with multimorbidity, prevalence estimates of polypharmacy (5+ drugs) and hyper-polypharmacy (10+ drugs) among persons aged 65 and older have increased significantly over the past 20 years [12,19] with recent international estimates ranging between 50–66% and 23–27%, respectively [20–23].

Exploring sex differences in multimorbidity and polypharmacy trends, and the interaction between these two health measures, is important given that men and women vary in key characterics (e.g., overall and disease-specific life expectancy, health behaviours, clinical presentation, health care use, level of caregiver/family support) that differentially shape their health status and medication use patterns as they age [24,25]. Estimates of multimorbidity are generally higher among women than men [2,8,9], though patterns vary with care setting [7], age groups, types of chronic conditions [26] and operational definitions examined [27]. Among older adults, some have reported comparable estimates of 2+ chronic conditions in women and men [4,8], but a higher prevalence of 5+ chronic conditions among older men [8]. Many [12,20–23], though not all [19], studies have also reported a higher prevalence of polypharmacy for older women. There are also notable sex differences in the prevalence of individual and co-occurring medication classes and chronic conditions [7,19] and potentially harmful drug-drug interactions [12]. Though recent studies have explored broad trends in multimorbidity [2,4,5] and/or polypharmacy [12,19,20,28], few have examined sex- and age-specific trends in both [20] among older adults or investigated variation over time in the impact of multimorbidity and age on medication use [29–32], particularly among older men and women.

We used linked administrative health data for adults aged ≥66 years in Ontario, Canada to examine sex differences in (i) the prevalence of multimorbidity, polypharmacy (5+ medications) and hyper-polypharmacy (10+ medications) at two time points (2003 and 2016), and (ii) the interactive associations between age, multimorbidity level, and time period on polypharmacy and hyper-polypharmacy measures. In addressing these objectives, we aim to demonstrate the impact of recent population-based trends in multimorbidity on changes in the complexity of medication use among specific age groups of older women and men.

## Methods

### Study design & sample

In this repeated cross-sectional study, we analyzed health administrative data for all Ontario residents aged ≥66 years with valid health insurance coverage on April 1, 2003 and/or April 1, 2016 (index dates). This age cut-off ensured all residents had a one-year lookback to assess medication coverage. Ontario is Canada's largest province by population and most residents are covered by a universal, publicly-funded health insurance program that pays for necessary medical and emergency services as well as prescription drugs for older (aged ≥65) adults. Over the study period there were no major changes in eligibility for these programs. The study is reported per RECORD guidelines (S1 Text) [33].

### Data sources & measures

Various health administrative databases from Ontario, Canada were used for this study (see S1 Table). These data were linked using unique encoded identifiers and analyzed at ICES. ICES (formerly known as the Institute for Clinical Evaluative Sciences) is an independent, non-profit research institute whose legal status under Ontario's health information privacy law allows it to collect and analyze health care and demographic data, without consent, for health system evaluation and improvement. Use of these data are authorized under section 45 of Ontario's Personal Health Information Protection Act, which does not require review by a Research Ethics Board.

**Multimorbidity.** For all persons identified, we captured the presence of 17 pre-specified common chronic conditions at the index dates from retrospective examination of hospital admissions (Discharge Abstract Database, DAD), physician claims (Ontario Health Insurance Program, OHIP) and drug dispensing (Ontario Drug Benefit [ODB] database) data. Included were acute myocardial infarction (AMI), asthma, (any) cancer, cardiac arrhythmia, chronic coronary syndrome, chronic obstructive pulmonary disease (COPD), congestive heart failure (CHF), dementia, diabetes, hypertension, non-psychotic mood and anxiety disorders, other mental illnesses (including schizophrenia, psychoses, personality disorders, substance abuse), osteoarthritis, osteoporosis, renal failure, rheumatoid arthritis, and stroke. These conditions were selected based on their population prevalence [6], system burden [10,34] and impact on quality of life, functional impairment and mortality of older adults and have been used in numerous prior studies of multimorbidity in Ontario and elsewhere [6,10,35–37]. Further, validated case ascertainment algorithms for administrative health data were available for many of these conditions (see S2 Table). We defined level of multimorbidity (i.e., disease burden) as a count of prevalent conditions coded as zero/one, two, three, four, or five-plus conditions.

**Polypharmacy outcomes.** We identified all outpatient prescriptions filled by residents in the 1-year before the index dates from the ODB data and derived the drug name and drug subclass through the drug identification number. We excluded records where the subclass was not identified, not applicable (e.g., contact lens solution), represented service fees, or where the drug name was a device. Polypharmacy was defined as having filled prescriptions for ≥5 unique drug names [20,23,38]. In secondary analyses, we quantified the presence of hyper-polypharmacy, defined as having filled prescriptions for ≥10 unique drug names [23,38]. The distribution of the most frequently dispensed drug subclasses among those with polypharmacy and hyper-polypharmacy was compared by sex at both index dates.

**Covariates.** Age and sex were derived from the Registered Persons Database (RPDB), and area-based income quintile and rural residence (community size <10,000 persons) were derived from the Canadian Census and Postal Code Conversion Files. We identified persons

residing in a long-term care home (LTC) based on having ≥1 claim(s) originating from LTC from the OHIP or ODB databases in the year prior to the index dates.

## Analysis

All descriptive and inferential statistics were stratified by sex. Descriptive analyses compared the distribution of characteristics in 2003 vs 2016. We modelled polypharmacy (or hyper-polypharmacy) using multivariable logistic regression analyses. Models were limited to records with complete information (<0.4% of all records were missing data on income or residence).

Given our focus on examining how time trends (2016 vs 2003 as reference) in polypharmacy varied according to age and level of multimorbidity, we incorporated a three-way interaction between year, age (continuous) and level of multimorbidity (0/1 to 5+ conditions) in these models. We first demonstrated the statistical significance of this interaction term using a Wald test (p<0.0001). We then included covariates for area-based income quintile and rural residence, two factors positively associated with higher medication use [23]. Further, we stratified these models by sex to illustrate variations in estimated associations among older women and men. From these models, we estimated average marginal probabilities of polypharmacy (or hyper-polypharmacy) in 2003 and in 2016 as well as risk ratios (RR) and risk differences (RD) comparing the change in the outcome (2016 vs 2003) at pre-specified ages (70, 80 and 90) and for each level of multimorbidity. All RRs and RDs were estimated using the method of marginal standardization [39], with 95% confidence intervals derived using the delta method. In sensitivity analyses we (i) excluded LTC residents from the models given their higher medication use, unique clinical characteristics, and varying patterns by sex [23], and (ii) conducted models for polypharmacy defined as ≥5 unique drug subclasses.

SAS Enterprise Guide v7.1 (SAS Institute Inc., Cary, NC) was used for data preparation and Stata MP v15.1 (StataCorp, College Station, TX) was used for all analyses.

## Results

Between 2003 and 2016, the prevalence of multimorbidity among persons aged ≥66 years increased from 75.9% to 82.1% for women and 72.5% to 79.4% for men (Table 1). Median age was higher among women than men in 2003 (75 vs 73) though this difference was slightly smaller in 2016. For both sexes, there was an increase in the relative proportion at higher income quintiles and residing in urban areas over time. Fig 1 illustrates the positive association between age and multimorbidity level and shift toward higher levels of multimorbidity over time among older women and men.

For both sexes, the largest absolute increases in the prevalence of select conditions were evident for diabetes, osteoarthritis, and hypertension whereas renal disease showed the largest relative increase over this period (Table 1). Though many conditions increased over time, there was a decrease in the prevalence of heart disease, non-psychotic mood/anxiety disorders, COPD and stroke for men and women. At both time points, prevalence estimates for cardiovascular conditions, diabetes, COPD and cancer were higher for men whereas women exhibited a higher prevalence of osteoarthritis, osteoporosis, non-psychotic mood/anxiety disorders and dementia.

In 2016, both men and women were dispensed a median of 6 (IQR: 3–10) prescription medications (this estimate had not changed for women but increased slightly for men relative to 2003). The prevalence of polypharmacy and hyper-polypharmacy increased over time and was higher among women than men at both time points. However, the absolute increase in both measures was greater among men than women (polypharmacy: from 55.7% in 2003 to 62.3% in 2016 for men vs 64.2% to 65.1% for women; hyper-polypharmacy: from 21.7% to

**Table 1. Characteristics of older adults (age ≥66 years) in Ontario, Canada: 2003 vs 2016 for women and men.**

| | Women | | Men | |
|---|---|---|---|---|
| | **2003** | **2016** | **2003** | **2016** |
| | **N = 829,533** | **N = 1,155,684** | **N = 622,647** | **N = 948,311** |
| *Sociodemographic Characteristics* | | | | |
| Age (Median, IQR) | 75 (70–81) | 74 (69–81) | 73 (69–78) | 73 (69–79) |
| 66–74 | 395,124 (47.6%) | 583,805 (50.5%) | 352,597 (56.6%) | 534,341 (56.3%) |
| 75–84 | 301,859 (36.4%) | 347,150 (30.0%) | 209,928 (33.7%) | 286,111 (30.2%) |
| 85+ | 132,550 (16.0%) | 224,729 (19.4%) | 60,122 (9.7%) | 127,859 (13.5%) |
| Income Quintile 1 (lowest) | 174,830 (21.1%) | 214,258 (18.5%) | 115,523 (18.6%) | 157,754 (16.6%) |
| Income Quintile 2 | 183,003 (22.1%) | 231,468 (20.0%) | 131,184 (21.1%) | 182,640 (19.3%) |
| Income Quintile 3 | 165,724 (20.0%) | 227,541 (19.7%) | 126,817 (20.4%) | 186,886 (19.7%) |
| Income Quintile 4 | 149,753 (18.1%) | 239,927 (20.8%) | 120,460 (19.3%) | 204,833 (21.6%) |
| Income Quintile 5 (highest) | 154,275 (18.6%) | 238,201 (20.6%) | 126,772 (20.4%) | 212,696 (22.4%) |
| Urban resident | 711,482 (85.8%) | 1,006,254 (87.1%) | 523,677 (84.1%) | 808,607 (85.3%) |
| Rural resident | 117,540 (14.2%) | 149,407 (12.9%) | 98,590 (15.8%) | 139,680 (14.7%) |
| Long term care resident flag (OHIP/ODB) | 45,518 (5.5%) | 53,549 (4.6%) | 17,112 (2.7%) | 22,726 (2.4%) |
| *Prevalent Chronic Conditions* | | | | |
| 0/1 Conditions | 199,503 (24.1%) | 207,275 (17.9%) | 171,093 (27.5%) | 195,692 (20.6%) |
| 2 Conditions | 191,753 (23.1%) | 241,970 (20.9%) | 137,095 (22.0%) | 194,401 (20.5%) |
| 3 Conditions | 173,885 (21.0%) | 253,792 (22.0%) | 122,184 (19.6%) | 196,626 (20.7%) |
| 4 Conditions | 120,388 (14.5%) | 193,968 (16.8%) | 86,073 (13.8%) | 150,940 (15.9%) |
| 5+ Conditions | 144,004 (17.4%) | 258,679 (22.4%) | 106,202 (17.1%) | 210,652 (22.2%) |
| **Multimorbidity (2+ Conditions)** | **630,030 (75.9%)** | **948,409 (82.1%)** | **451,554 (72.5%)** | **752,619 (79.4%)** |
| # conditions (Median, IQR) | 3 (2–4) | 3 (2–4) | 3 (1–4) | 3 (2–4) |
| Acute myocardial infarction (AMI) | 4,431 (0.5%) | 3,656 (0.3%) | 5,385 (0.9%) | 5,034 (0.5%) |
| Cardiac arrythmia | 98,523 (11.9%) | 158,080 (13.7%) | 85,309 (13.7%) | 148,228 (15.6%) |
| Asthma | 97,841 (11.8%) | 174,892 (15.1%) | 60,568 (9.7%) | 99,984 (10.5%) |
| Cancer | 131,198 (15.8%) | 223,823 (19.4%) | 137,551 (22.1%) | 239,946 (25.3%) |
| Congestive heart failure (CHF) | 87,002 (10.5%) | 100,907 (8.7%) | 69,833 (11.2%) | 97,422 (10.3%) |
| Chronic obstructive pulmonary disease (COPD) | 73,060 (8.8%) | 96,137 (8.3%) | 78,709 (12.6%) | 94,422 (10.0%) |
| Chronic coronary syndrome | 218,203 (26.3%) | 245,105 (21.2%) | 215,627 (34.6%) | 306,358 (32.3%) |
| Dementia | 61,884 (7.5%) | 105,222 (9.1%) | 30,249 (4.9%) | 61,664 (6.5%) |
| Diabetes | 147,553 (17.8%) | 323,678 (28.0%) | 139,143 (22.3%) | 329,330 (34.7%) |
| Hypertension | 545,622 (65.8%) | 823,045 (71.2%) | 370,993 (59.6%) | 666,251 (70.3%) |
| Non-psychotic Mood/Anxiety Disorders | 148,925 (18.0%) | 150,853 (13.1%) | 74,329 (11.9%) | 79,845 (8.4%) |
| (Other) Mental Health Conditions | 28,034 (3.4%) | 44,447 (3.8%) | 28,118 (4.5%) | 47,453 (5.0%) |
| Osteoarthritis | 503,067 (60.6%) | 816,641 (70.7%) | 328,934 (52.8%) | 590,424 (62.3%) |
| Osteoporosis | 128,047 (15.4%) | 257,871 (22.3%) | 14,458 (2.3%) | 35,173 (3.7%) |
| Renal Disease | 20,603 (2.5%) | 83,180 (7.2%) | 25,067 (4.0%) | 89,591 (9.4%) |
| Rheumatoid Arthritis | 19,720 (2.4%) | 37,586 (3.3%) | 7,704 (1.2%) | 15,961 (1.7%) |
| Stroke | 52,460 (6.3%) | 65,603 (5.7%) | 47,378 (7.6%) | 67,005 (7.1%) |
| *Drug Outcomes* | | | | |
| # unique drug names dispensed (Median, IQR) | 6 (3–10) | 6 (3–10) | 5 (2–9) | 6 (3–10) |
| 0 | 67,854 (8.2%) | 80,892 (7.0%) | 75,755 (12.2%) | 82,578 (8.7%) |
| 1 | 41,866 (5.0%) | 62,309 (5.4%) | 43,590 (7.0%) | 54,478 (5.7%) |
| 2 | 54,813 (6.6%) | 77,169 (6.7%) | 50,047 (8.0%) | 65,409 (6.9%) |
| 3 | 63,757 (7.7%) | 88,632 (7.7%) | 52,928 (8.5%) | 74,573 (7.9%) |
| 4 | 68,869 (8.3%) | 94,472 (8.2%) | 53,476 (8.6%) | 80,090 (8.4%) |

*(Continued)*

**Table 1.** (Continued)

| | Women | | Men | |
|---|---|---|---|---|
| | **2003** | **2016** | **2003** | **2016** |
| | **N = 829,533** | **N = 1,155,684** | **N = 622,647** | **N = 948,311** |
| 5+ | 532,374 (64.2%) | 752,210 (65.1%) | 346,851 (55.7%) | 591,183 (62.3%) |
| **Polypharmacy (5+ drug names)** | **532,374 (64.2%)** | **752,210 (65.1%)** | **346,851 (55.7%)** | **591,183 (62.3%)** |
| **Hyper-polypharmacy (10+ drug names)** | **230,126 (27.7%)** | **335,045 (29.0%)** | **135,149 (21.7%)** | **247,701 (26.1%)** |
| # drug subclasses dispensed (Median, IQR) | 6 (3–9) | 6 (3–10) | 5 (2–8) | 6 (3–9) |
| **Polypharmacy (5+ drug subclasses)** | **514,580 (62.0%)** | **735,511 (63.6%)** | **330,834 (53.1%)** | **570,586 (60.2%)** |
| **Hyper-polypharmacy (10+ drug subclasses)** | **190,284 (22.9%)** | **291,644 (25.2%)** | **107,741 (17.3%)** | **205,158 (21.6%)** |

**Notes:** N (column %) shown unless otherwise stated.

IQR = interquartile range; OHIP = Ontario Health Insurance Program; ODB = Ontario Drug Benefit database.

26.1% for men vs 27.7% to 29.0% for women). The increase in prevalence of polypharmacy with age and over time among women and men is shown in Fig 2.

The distribution of total population characteristics for older adults in 2003 and 2016 not stratified by sex are presented in S3 Table.

Sex differences in the marginal probabilities of polypharmacy in 2016 vs 2003 according to pre-specified ages and multimorbidity level, and corresponding unadjusted and adjusted (for income quintile and rural residence) risk differences and risk ratios for change in polypharmacy estimates (2016 vs 2003) are presented in Table 2 (see S4 Table for model parameter estimates). At both time points, polypharmacy increased significantly with each level of multimorbidity (0/1 up to 5+ conditions) for women and men of all ages. The estimated change in polypharmacy prevalence (2016 vs 2003) by multimorbidity level was age and sex dependent. For example, there was a decrease in polypharmacy over time for women aged < 90 years and this decrease (both in absolute and relative terms) was generally more pronounced among women at the youngest age (age 70) and with lower levels of multimorbidity (e.g., for women at age 70, the adjusted risk differences and risk ratios capturing absolute and relative change in polypharmacy were -3.5 and 0.87 [those with no or 1 condition] vs. -2.4 and 0.97 [those with 5 + conditions]). For women at age 90 and above, there was a small increase (statistically significant for some multimorbidity levels) in polypharmacy over time. Conversely, between 2003 and 2016, polypharmacy increased among men at all ages and multimorbidity levels. The relative increase was most pronounced among men at the younger ages and with lower levels of multimorbidity (e.g., for men at age 70, the adjusted risk ratios capturing relative change in polypharmacy were 1.11 [for those with no or 1 condition] vs 1.01 [for those with 5+ conditions]). The absolute increase was generally larger for men at the older ages (e.g., 90+) and among those with 4 or fewer conditions. The sex difference in polypharmacy estimates (e.g., higher in women than men) was also smaller in 2016 than 2003.

Generally comparable findings for women and men were observed for regression models estimating change in hyper-polypharmacy by age and multimorbidity level (Table 3), though there was some variation in the magnitude of absolute and relative differences (for selected age-multimorbidity levels).

Our findings also remained robust in our sensitivity analyses which excluded those residing in LTC (representing <5% of the total study sample) (see S5 Table) and modeled polypharmacy defined as ≥5 unique drug subclasses (see S6 Table).

The 10 most common prescription drug subclasses dispensed among women and men using 5+ or 10+ medications in 2016, along with absolute and percent changes relative to 2003, are presented in Table 4. Among both groups, estimates for selected cardiovascular (statins,

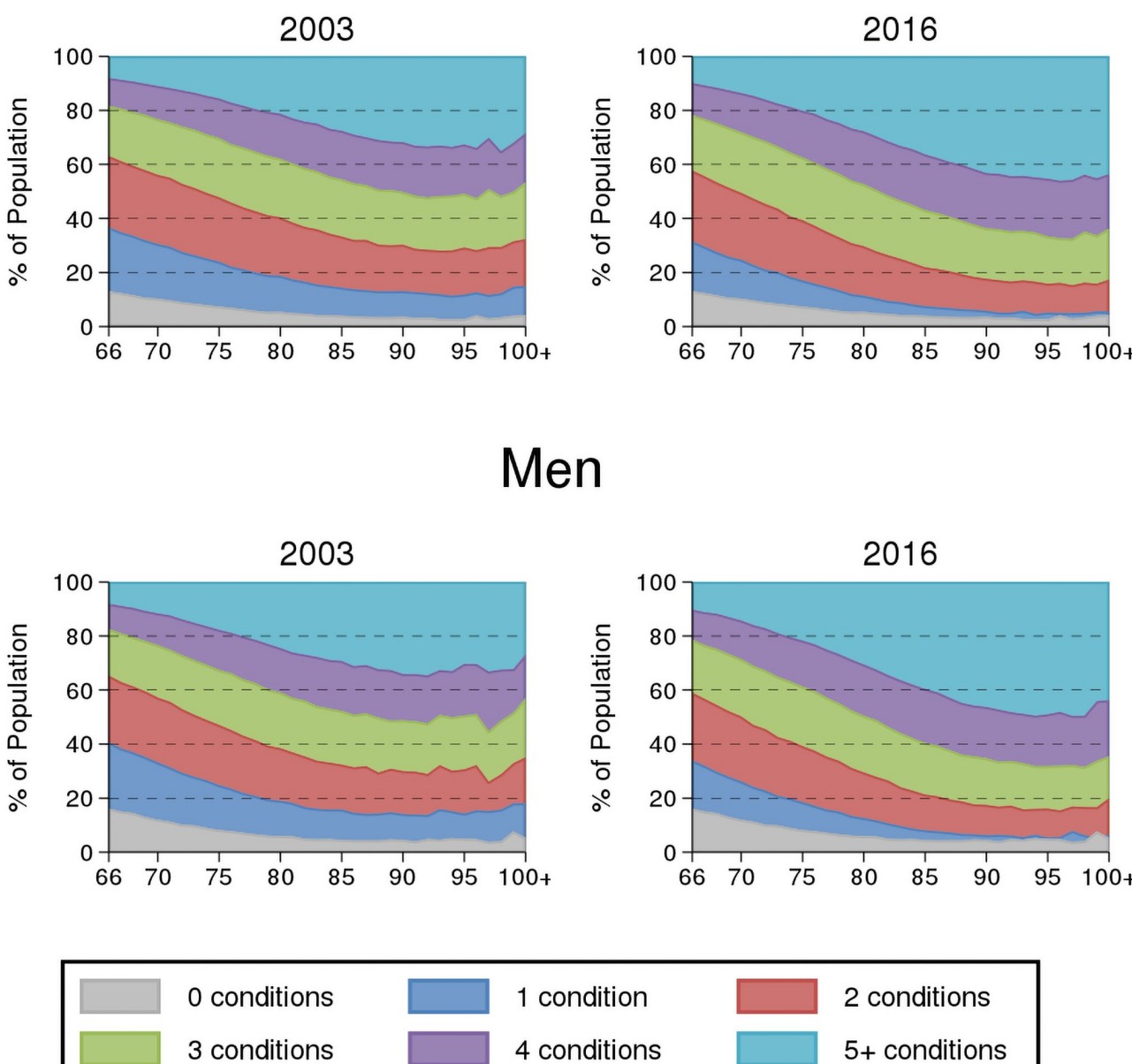

**Fig 1. Distribution of level of multimorbidity by age among older adults (age ≥66 years) in Ontario, Canada: 2003 vs 2016 for women and men.**

angiotensin-converting enzyme (ACE) inhibitors, beta-blockers) and oral anti-diabetic medications were higher among men than women (likely reflecting sex differences in the prevalence of related conditions) whereas women were more often dispensed proton pump inhibitors (PPIs) and medications for thyroid disease. Medications for prostate enlargement were among the top 10 subclasses for men with polypharmacy and hyper-polypharmacy in 2016 and showed the largest percent increase over the 13 years. Relative to 2003 (see S7 Table for 10 most common subclasses in 2003), there was a notable rise in statin and PPI use and decline in

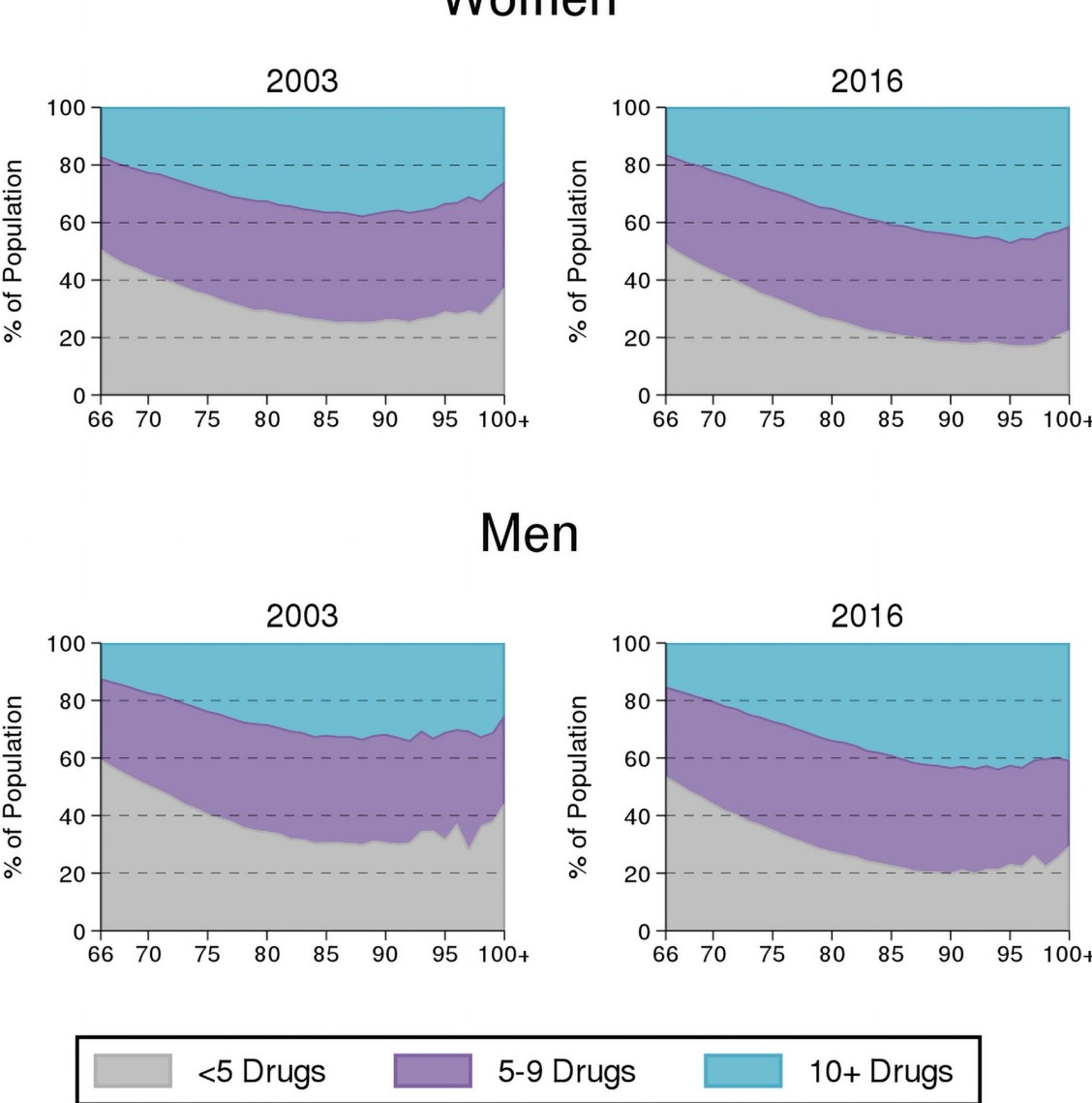

**Fig 2. Distribution of polypharmacy and hyper-polypharmacy by age among older adults (age ≥66 years) in Ontario, Canada: 2003 vs 2016 for women and men.**

use of non-steroidal anti-inflammatory drugs (NSAIDs) and benzodiazepines (BZDs) among men and women with polypharmacy and hyper-polypharmacy. There was also a decline in the use of bisphosphonates in women and coronary vasodilators in men. Among both sexes with hyper-polypharmacy, significant proportions were receiving statins (76% men, 64% women), PPIs (54% men, 61% women) and corticosteroids (52% men, 53% women) in 2016.

## Discussion

### Main findings

In this population-based, repeated cross-sectional study of Ontarians aged ≥66 years, we showed a statistically significant increase in the prevalence of multimorbidity, polypharmacy

**Table 2. Marginal probabilities of polypharmacy (5+ drugs) and unadjusted and adjusted risk differences and risk ratios (95% Confidence Intervals), by select ages, level of multimorbidity and sex.**

| Women | | | | | | | |
|---|---|---|---|---|---|---|---|
| Age | Level of MMB | P(2003) | P(2016) | Unadj Risk Difference (95% CI) | Adj Risk Difference (95% CI) | Unadj Risk Ratio (95% CI) | Adj Risk Ratio (95% CI) |
| 70 | 0/1 | 25.9 | 22.2 | -3.73 (-4.03, -3.43)* | -3.50 (-3.80, -3.20)* | 0.86 (0.85, 0.87)* | 0.87 (0.85, 0.88)* |
| 70 | 2 | 55.2 | 48.0 | -7.24 (-7.61, -6.88)* | -6.92 (-7.29, -6.56)* | 0.87 (0.86, 0.87)* | 0.87 (0.87, 0.88)* |
| 70 | 3 | 72.8 | 66.8 | -6.02 (-6.38, -5.66)* | -5.78 (-6.14, -5.42)* | 0.92 (0.91, 0.92)* | 0.92 (0.92, 0.93)* |
| 70 | 4 | 84.8 | 80.2 | -4.62 (-4.99, -4.24)* | -4.48 (-4.85, -4.10)* | 0.95 (0.94, 0.95)* | 0.95 (0.94, 0.95)* |
| 70 | 5+ | 94.4 | 92.0 | -2.41 (-2.66, -2.16)* | -2.39 (-2.64, -2.14)* | 0.97 (0.97, 0.98)* | 0.97 (0.97, 0.98)* |
| 80 | 0/1 | 31.4 | 29.7 | -1.78 (-2.19, -1.37)* | -1.51 (-1.93, -1.10)* | 0.94 (0.93, 0.96)* | 0.95 (0.94, 0.96)* |
| 80 | 2 | 59.6 | 56.1 | -3.51 (-3.88, -3.13)* | -3.23 (-3.61, -2.86)* | 0.94 (0.93, 0.95)* | 0.95 (0.94, 0.95)* |
| 80 | 3 | 75.0 | 72.4 | -2.58 (-2.89, -2.26)* | -2.38 (-2.69, -2.07)* | 0.97 (0.96, 0.97)* | 0.97 (0.96, 0.97)* |
| 80 | 4 | 85.2 | 83.2 | -1.97 (-2.25, -1.68)* | -1.83 (-2.12, -1.55)* | 0.98 (0.97, 0.98)* | 0.98 (0.98, 0.98)* |
| 80 | 5+ | 93.6 | 93.1 | -0.50 (-0.66, -0.34)* | -0.46 (-0.63, -0.30)* | 0.99 (0.99, 1.00)* | 1.00 (0.99, 1.00)* |
| 90 | 0/1 | 37.5 | 38.4 | 0.86 (-0.02, 1.74) | 1.14 (0.26, 2.02)* | 1.02 (1.00, 1.05) | 1.03 (1.01, 1.05)* |
| 90 | 2 | 63.8 | 63.9 | 0.06 (-0.63, 0.76) | 0.29 (-0.40, 0.98) | 1.00 (0.99, 1.01) | 1.00 (0.99, 1.02) |
| 90 | 3 | 77.1 | 77.4 | 0.35 (-0.19, 0.90) | 0.52 (-0.03, 1.07) | 1.00 (1.00, 1.01) | 1.01 (1.00, 1.01) |
| 90 | 4 | 85.6 | 85.9 | 0.31 (-0.18, 0.80) | 0.44 (-0.06, 0.93) | 1.00 (1.00, 1.01) | 1.01 (1.00, 1.01) |
| 90 | 5+ | 92.8 | 94.1 | 1.34 (1.06, 1.62)* | 1.39 (1.11, 1.67)* | 1.01 (1.01, 1.02)* | 1.01 (1.01, 1.02)* |
| Men | | | | | | | |
| Age | Level of MMB | P(2003) | P(2016) | Unadj Risk Difference (95% CI) | Adj Risk Difference (95% CI) | Unadj Risk Ratio (95% CI) | Adj Risk Ratio (95% CI) |
| 70 | 0/1 | 17.6 | 19.4 | 1.81 (1.53, 2.08)* | 1.92 (1.65, 2.20)* | 1.10 (1.09, 1.12)* | 1.11 (1.09, 1.13)* |
| 70 | 2 | 45.1 | 47.1 | 2.08 (1.68, 2.48)* | 2.28 (1.88, 2.68)* | 1.05 (1.04, 1.06)* | 1.05 (1.04, 1.06)* |
| 70 | 3 | 65.2 | 66.9 | 1.67 (1.26, 2.09)* | 1.84 (1.42, 2.26)* | 1.03 (1.02, 1.03)* | 1.03 (1.02, 1.03)* |
| 70 | 4 | 79.6 | 81.0 | 1.37 (0.93, 1.81)* | 1.48 (1.04, 1.92)* | 1.02 (1.01, 1.02)* | 1.02 (1.01, 1.02)* |
| 70 | 5+ | 92.1 | 92.6 | 0.47 (0.19, 0.76)* | 0.51 (0.23, 0.80)* | 1.01 (1.00, 1.01)* | 1.01 (1.00, 1.01)* |
| 80 | 0/1 | 24.2 | 26.2 | 2.05 (1.58, 2.53)* | 2.21 (1.74, 2.69)* | 1.08 (1.06, 1.11)* | 1.09 (1.07, 1.11)* |
| 80 | 2 | 51.7 | 53.7 | 1.99 (1.48, 2.50)* | 2.21 (1.71, 2.72)* | 1.04 (1.03, 1.05)* | 1.04 (1.03, 1.05)* |
| 80 | 3 | 69.1 | 71.2 | 2.11 (1.67, 2.54)* | 2.27 (1.83, 2.70)* | 1.03 (1.02, 1.04)* | 1.03 (1.03, 1.04)* |
| 80 | 4 | 80.9 | 82.8 | 1.94 (1.55, 2.33)* | 2.05 (1.66, 2.44)* | 1.02 (1.02, 1.03)* | 1.03 (1.02, 1.03)* |
| 80 | 5+ | 91.9 | 93.0 | 1.11 (0.90, 1.32)* | 1.15 (0.94, 1.36)* | 1.01 (1.01, 1.01)* | 1.01 (1.01, 1.01)* |
| 90 | 0/1 | 32.3 | 34.5 | 2.18 (1.09, 3.26)* | 2.38 (1.30, 3.47)* | 1.07 (1.03, 1.10)* | 1.07 (1.04, 1.11)* |
| 90 | 2 | 58.2 | 60.1 | 1.83 (0.86, 2.81)* | 2.08 (1.10, 3.05)* | 1.03 (1.01, 1.05)* | 1.04 (1.02, 1.05)* |
| 90 | 3 | 72.7 | 75.1 | 2.44 (1.65, 3.24)* | 2.60 (1.80, 3.39)* | 1.03 (1.02, 1.04)* | 1.04 (1.02, 1.05)* |
| 90 | 4 | 82.1 | 84.5 | 2.42 (1.69, 3.15)* | 2.54 (1.80, 3.27)* | 1.03 (1.02, 1.04)* | 1.03 (1.02, 1.04)* |
| 90 | 5+ | 91.6 | 93.3 | 1.74 (1.32, 2.15)* | 1.78 (1.36, 2.20)* | 1.02 (1.01, 1.02)* | 1.02 (1.01, 1.02)* |

**Notes:** P(2003) and P(2016) correspond to predicted probabilities of the outcome (polypharmacy) from unadjusted logistic regression.

Level of MMB = level of multimorbidity (number of conditions); RD = risk (prevalence) difference; RR = risk (prevalence) ratio; Unadj = unadjusted; Adj = adjusted (for rurality and area-based income quintile).

All 95% confidence intervals are calculated using the delta method (* denotes $p < 0.05$).

and hyper-polypharmacy over a 13-year period. By 2016, approximately two thirds of all older adults were receiving 5+ prescription medications and just over one quarter were prescribed 10+ medications. Though prevalence estimates of multimorbidity and polypharmacy were higher for women than men in both 2003 and 2016, the absolute increase in these measures over time was greater for men (especially for polypharmacy and hyper-polypharmacy) leading to a narrowing of the gap between the sexes by 2016. Importantly, we observed significant age and sex differences in the association between multimorbidity level and change in polypharmacy outcomes. Whereas polypharmacy decreased over this period among women aged < 90

**Table 3. Marginal probabilities of hyper-polypharmacy (10+ drugs) and unadjusted and adjusted risk differences and risk ratios (95% Confidence Intervals), by select ages, level of multimorbidity and sex.**

| Age | Level of MMB | P(2003) | P(2016) | Unadj Risk Difference (95% CI) | Adj Risk Difference (95% CI) | Unadj Risk Ratio (95% CI) | Adj Risk Ratio (95% CI) |
|---|---|---|---|---|---|---|---|
| **Women** | | | | | | | |
| 70 | 0/1 | 4.0 | 2.9 | -1.10 (-1.23, -0.97)* | -1.05 (-1.18, -0.92)* | 0.73 (0.70, 0.75)* | 0.74 (0.71, 0.77)* |
| 70 | 2 | 13.5 | 10.2 | -3.30 (-3.53, -3.06)* | -3.15 (-3.39, -2.91)* | 0.76 (0.74, 0.77)* | 0.77 (0.75, 0.78)* |
| 70 | 3 | 26.3 | 21.6 | -4.67 (-5.01, -4.33)* | -4.42 (-4.76, -4.08)* | 0.82 (0.81, 0.83)* | 0.83 (0.82, 0.84)* |
| 70 | 4 | 41.8 | 36.1 | -5.73 (-6.22, -5.24)* | -5.42 (-5.91, -4.93)* | 0.86 (0.85, 0.87)* | 0.87 (0.86, 0.88)* |
| 70 | 5+ | 67.4 | 62.1 | -5.32 (-5.80, -4.84)* | -5.14 (-5.62, -4.67)* | 0.92 (0.91, 0.93)* | 0.92 (0.92, 0.93)* |
| 80 | 0/1 | 5.6 | 4.7 | -0.87 (-1.06, -0.68)* | -0.79 (-0.98, -0.60)* | 0.84 (0.81, 0.87)* | 0.86 (0.83, 0.89)* |
| 80 | 2 | 15.8 | 13.7 | -2.14 (-2.40, -1.88)* | -1.96 (-2.22, -1.70)* | 0.86 (0.85, 0.88)* | 0.88 (0.86, 0.89)* |
| 80 | 3 | 28.3 | 25.4 | -2.94 (-3.25, -2.63)* | -2.68 (-2.99, -2.38)* | 0.90 (0.89, 0.91)* | 0.90 (0.89, 0.91)* |
| 80 | 4 | 42.1 | 39.0 | -3.08 (-3.45, -2.70)* | -2.77 (-3.15, -2.40)* | 0.93 (0.92, 0.94)* | 0.93 (0.92, 0.94)* |
| 80 | 5+ | 65.2 | 63.6 | -1.53 (-1.84, -1.22)* | -1.36 (-1.67, -1.05)* | 0.98 (0.97, 0.98)* | 0.98 (0.97, 0.98)* |
| 90 | 0/1 | 7.7 | 7.5 | -0.18 (-0.69, 0.33) | -0.07 (-0.58, 0.44) | 0.98 (0.91, 1.04) | 0.99 (0.92, 1.06) |
| 90 | 2 | 18.5 | 18.2 | -0.34 (-0.93, 0.24) | -0.15 (-0.73, 0.43) | 0.98 (0.95, 1.01) | 0.99 (0.96, 1.02) |
| 90 | 3 | 30.5 | 29.6 | -0.89 (-1.51, -0.27)* | -0.64 (-1.26, -0.03)* | 0.97 (0.95, 0.99)* | 0.98 (0.96, 1.00)* |
| 90 | 4 | 42.4 | 42.0 | -0.34 (-1.04, 0.36) | -0.05 (-0.75, 0.64) | 0.99 (0.98, 1.01) | 1.00 (0.98, 1.02) |
| 90 | 5+ | 62.8 | 65.1 | 2.30 (1.77, 2.84)* | 2.46 (1.93, 3.00)* | 1.04 (1.03, 1.05)* | 1.04 (1.03, 1.05)* |
| **Men** | | | | | | | |
| 70 | 0/1 | 2.3 | 2.3 | 0.06 (-0.04, 0.17) | 0.09 (-0.02, 0.20) | 1.03 (0.98, 1.08) | 1.04 (0.99, 1.09) |
| 70 | 2 | 8.7 | 8.9 | 0.23 (0.01, 0.46)* | 0.34 (0.11, 0.57)* | 1.03 (1.00, 1.05)* | 1.04 (1.01, 1.07)* |
| 70 | 3 | 18.7 | 19.3 | 0.60 (0.25, 0.94)* | 0.79 (0.44, 1.13)* | 1.03 (1.01, 1.05)* | 1.04 (1.02, 1.06)* |
| 70 | 4 | 32.7 | 33.4 | 0.75 (0.23, 1.26)* | 0.98 (0.47, 1.50)* | 1.02 (1.01, 1.04)* | 1.03 (1.01, 1.05)* |
| 70 | 5+ | 58.4 | 59.9 | 1.43 (0.90, 1.96)* | 1.64 (1.11, 2.17)* | 1.02 (1.02, 1.03)* | 1.03 (1.02, 1.04)* |
| 80 | 0/1 | 3.8 | 3.9 | 0.16 (-0.04, 0.36) | 0.21 (0.01, 0.41)* | 1.04 (0.99, 1.10) | 1.06 (1.00, 1.11)* |
| 80 | 2 | 11.8 | 11.9 | 0.09 (-0.22, 0.41) | 0.24 (-0.07, 0.56) | 1.01 (0.98, 1.03) | 1.02 (0.99, 1.05) |
| 80 | 3 | 22.3 | 22.7 | 0.34 (-0.05, 0.72) | 0.55 (0.16, 0.93)* | 1.02 (1.00, 1.03) | 1.02 (1.01, 1.04)* |
| 80 | 4 | 35.0 | 36.0 | 1.02 (0.54, 1.49)* | 1.28 (0.80, 1.75)* | 1.03 (1.02, 1.04)* | 1.04 (1.02, 1.05)* |
| 80 | 5+ | 59.4 | 61.7 | 2.27 (1.88, 2.66)* | 2.46 (2.07, 2.85)* | 1.04 (1.03, 1.04)* | 1.04 (1.03, 1.05)* |
| 90 | 0/1 | 6.2 | 6.5 | 0.34 (-0.28, 0.97) | 0.43 (-0.19, 1.05) | 1.06 (0.95, 1.16) | 1.07 (0.97, 1.17) |
| 90 | 2 | 15.9 | 15.8 | -0.15 (-0.94, 0.65) | 0.06 (-0.72, 0.85) | 0.99 (0.94, 1.04) | 1.00 (0.95, 1.05) |
| 90 | 3 | 26.5 | 26.5 | -0.01 (-0.86, 0.84) | 0.22 (-0.62, 1.07) | 1.00 (0.97, 1.03) | 1.01 (0.98, 1.04) |
| 90 | 4 | 37.3 | 38.6 | 1.30 (0.33, 2.27)* | 1.58 (0.62, 2.54)* | 1.03 (1.01, 1.06)* | 1.04 (1.02, 1.07)* |
| 90 | 5+ | 60.4 | 63.5 | 3.09 (2.34, 3.83)* | 3.26 (2.52, 4.00)* | 1.05 (1.04, 1.06)* | 1.05 (1.04, 1.07)* |

**Notes:** P(2003) and P(2016) correspond to predicted probabilities of the outcome (hyper-polypharmacy) from unadjusted logistic regression.

Level of MMB = level of multimorbidity (number of conditions); RD = risk (prevalence) difference; RR = risk (prevalence) ratio; Unadj = unadjusted; Adj = adjusted (for rurality and area-based income quintile).

All 95% confidence intervals are calculated using the delta method (* denotes p<0.05).

years (especially for relatively younger women and those with fewer conditions), it increased among men at all ages and multimorbidity levels (with larger absolute increases typically at older ages and among those with 4 or fewer conditions).

## What was known previously and what does our study adds

Population-based investigations of sex- and age-specific trends in *both* multimorbidity and polypharmacy are scarce [20] especially among persons aged 65 and older. Additionally,

**Table 4. Most frequently dispensed drug subclasses for women and men (age ≥66 years) in Ontario, Canada in 2016 with polypharmacy and hyper-polypharmacy, and absolute/percent change from 2003.**

| Women | Polypharmacy | | | | Hyper-polypharmacy | | |
|---|---|---|---|---|---|---|---|
| | 2016 Prevalence N = 752,210 | Absolute change from 2003 | Percent change from 2003 | | 2016 Prevalence N = 335,045 | Absolute change from 2003 | Percent change from 2003 |
| ANTILIPEMIC: STATINS | 56.5% | +23.0 | 68.5% | ANTILIPEMIC: STATINS | 63.9% | +25.1 | 64.5% |
| PROTON PUMP INHIBITORS | 45.4% | +23.9 | 111.2% | PROTON PUMP INHIBITORS | 60.8% | +28.7 | 89.5% |
| CORTICOSTEROIDS, PLAIN[a] | 39.8% | -2.0 | -4.8% | CORTICOSTEROIDS, PLAIN[a] | 52.6% | -3.0 | -5.4% |
| CALCIUM BLOCKERS | 34.9% | +1.4 | 4.1% | CALCIUM BLOCKERS | 43.5% | +0.6 | 1.3% |
| DIURETICS | 32.1% | -7.6 | -19.2% | DIURETICS | 42.4% | -8.1 | -16.1% |
| BETA-BLOCKERS | 30.7% | +0.5 | 1.8% | NARCOTICS: OPIATE AGONISTS | 41.9% | -3.8 | -8.3% |
| NARCOTICS: OPIATE AGONISTS | 28.1% | -3.7 | -11.7% | BETA-BLOCKERS | 40.1% | +3.3 | 9.0% |
| ACE INHIBITORS | 28.0% | -10.0 | -26.3% | ACE INHIBITORS | 32.0% | -13.4 | -29.5% |
| HYPOTHYROIDISM THERAPY | 26.5% | +3.5 | 15.1% | CATHARTICS AND LAXATIVES | 31.8% | +3.9 | 14.2% |
| ANGIOTENSIN II ANTAGONIST | 23.2% | +12.0 | 108.0% | HYPOTHYROIDISM THERAPY | 30.9% | +4.1 | 15.5% |
| Men | Polypharmacy | | | | Hyper-polypharmacy | | |
| | 2016 Prevalence N = 591,183 | Absolute change from 2003 | Percent change from 2003 | | 2016 Prevalence N = 247,701 | Absolute change from 2003 | Percent change from 2003 |
| ANTILIPEMIC: STATINS | 70.3% | +26.8 | 61.6% | ANTILIPEMIC: STATINS | 76.3% | +27.2 | 55.6% |
| ACE INHIBITORS | 38.9% | -9.3 | -19.2% | PROTON PUMP INHIBITORS | 53.6% | +26.1 | 94.8% |
| PROTON PUMP INHIBITORS | 38.8% | +20.3 | 109.6% | CORTICOSTEROIDS, PLAIN[a] | 51.7% | -5.6 | -9.7% |
| CORTICOSTEROIDS, PLAIN[a] | 37.8% | -4.6 | -10.9% | BETA-BLOCKERS | 47.8% | +5.2 | 12.3% |
| BETA-BLOCKERS | 37.1% | +1.6 | 4.4% | ACE INHIBITORS | 42.5% | -13.5 | -24.1% |
| CALCIUM BLOCKERS | 33.2% | +0.2 | 0.6% | CALCIUM BLOCKERS | 41.0% | -0.5 | -1.2% |
| ORAL ANTI-GLYCEMICS | 30.4% | +11.2 | 58.3% | ORAL ANTI-GLYCEMICS | 39.7% | +14.4 | 57.0% |
| DIURETICS | 27.5% | -6.7 | -19.5% | NARCOTICS: OPIATE AGONISTS | 39.3% | -5.2 | -11.6% |
| PROSTATIC HYPERPLASIA | 27.3% | +23.3 | 577.5% | DIURETICS | 39.1% | -7.5 | -16.1% |
| NARCOTICS: OPIATE AGONISTS | 26.5% | -4.7 | -15.1% | PROSTATIC HYPERPLASIA | 36.0% | +30.2 | 513.9% |

Notes:

[a] includes systemic products (e.g., oral) and products for local use.

among studies that have examined the independent association between multimorbidity and polypharmacy, most have focused on relatively younger populations [29–31] or were limited to older adults receiving home [21] or hospital [32] care or older data sources [19]. Consistent with previous observations of a strong positive association between number of chronic conditions and polypharmacy [19–22,31], our findings reveal that at any given age among men and women aged greater than 65 years, the prevalence of polypharmacy and hyper-polypharmacy is driven largely by the level of multimorbidity. Further, among those with 5+ chronic

conditions, the prevalence of polypharmacy is high regardless of age (e.g., polypharmacy and hyper-polypharmacy estimates among those aged 70, 80 and 90 years, ranged between 92–94% and 60–65%, respectively). For many older adults living with multimorbidity, a high rate of use of multiple medications may reflect appropriate pharmacotherapeutic care [40,41]. However, the concurrent use of multiple medications raises concerns about the potential of drug-drug and drug-disease interactions, adverse drug events, treatment burden and non-adherence among this population [12–19,23,41]. The relative balance between benefit and harm is a priority area for further clinical and research investigations.

The overall trends we observed in multimorbidity and in specific chronic conditions over the 13 years parallel many of those observed [4,5,20] and/or projected [1] by other investigators. Previous studies have also shown significant increases over time in polypharmacy and hyper-polypharmacy among older adults [12,19,20,23,28,42], though the magnitude of this increase appears to be diminishing, with a stabilization or even slight decrease noted, over more recent time periods in some regions [23,28,43,44]. Overall, our prevalence estimates of multimorbidity and polypharmacy in 2016 are generally consistent with (albeit higher than some) estimates reported for North America and Europe during comparable time periods [20–23,44].

Consistent with several recent studies [4,19,20,42], we showed higher absolute and relative increases in the prevalence of multimorbidity and polypharmacy over time among older men as compared to women even though prevalence estimates remained higher for women than men. There may be several factors contributing to this narrowing of sex differences in multi-morbidity and polypharmacy prevalence among successive cohorts of older adults, including variation between the sexes in gains in overall life expectancy (with higher absolute increases observed among older men than women [45]), health behaviours and healthcare use at earlier ages [24,25]. As our study design and data do not permit a rigorous exploration of these hypotheses, they remain important areas for further research. However, our findings also indicated that the increases observed in polypharmacy over time in relation to multimorbidity level were not uniform across age and sex within our older population. Whereas polypharmacy decreased among women aged <90 (especially for those at the youngest ages and with fewer conditions), it increased among older men (with larger absolute increases typically evident for those aged 80–90+ years and with 4 or fewer conditions). A U.S. study of polypharmacy trends between 1988 and 2010 among national survey respondents aged 65 and older also showed a more pronounced increase in polypharmacy among males primarily for those aged 80+ years [19]. Our finding of a decline in polypharmacy among older women may help explain the relatively lower prevalence and recent decline in polypharmacy and potentially inappropriate medication measures reported for the oldest old [44] and those in long-term care settings [23], as over two-thirds of these populations are women. These recent positive trends for older women [19] may reflect the relatively greater research and clinical attention historically directed to reducing potentially high-risk medications among older women because of their consistently higher use of multiple and potentially inappropriate medications relative to men [23,46].

We do not believe the disproportionate increase in polypharmacy among older men, especially evident for those at younger ages and with fewer conditions, is due to the modestly higher increase in multimorbidity we observed in older men compared to women. The sex difference in polypharmacy increase was over 8-fold higher than that shown for multimorbidity. Other relevant drivers of increasing polypharmacy among older men could include: more timely diagnosis and treatment of chronic conditions; earlier detection and treatment of mild symptoms; greater availability, access and/or acceptance of effective preventive therapies especially at younger ages; targeted pharmaceutical promotion (for male-related health concerns leading to greater demand from patients); increase in particular clusters of conditions associated with multiple medication use; changes in patient healthcare utilization, expectations and/

or coverage; changes in medical practice (e.g., indications for treatment, use of multiple agents for single conditions); and, other unique patient, prescriber and health system factors [19,22,42,44]. Though there were differences in the distribution of common medication sub-classes among women and men with prevalent polypharmacy and hyper-polypharmacy, both sexes showed increases in statin and PPI use and decreases in NSAID and BZD use over this period consistent with other recent Canadian [23] and European [28,42] reports.

## Strengths and limitations

Our study has several strengths including the use of population-based data (capturing all persons aged ≥66 in both community and institutional settings), exploration of trends in both multimorbidity and polypharmacy (and for the latter, over a more contemporary time period than previous studies [19,42–44]), use of validated methods and algorithms for our multimorbidity measures, and novel exploration of sex differences in the association between multimorbidity level, age and time and polypharmacy and hyper-polypharmacy outcomes. Relevant limitations include the observational cross-sectional study design (limiting any conclusions about the causal nature of observed associations), use of secondary health administrative data that may pose risks for misclassification bias in our study measures, and use of prescription claims data which may not necessarily reflect actual medication use and excludes non-prescription (i.e., over-the-counter and natural health product) drug use. Our multimorbidity estimates reflect the chronic conditions we examined and may vary from studies including more or different conditions. Changes over time in the detection, diagnosis and coding of selected chronic conditions in administrative health databases [47] as well as in the prescription (vs. non-prescription) status of medications and/or their coverage in the publicly funded drug benefit insurance plan may represents potential sources of bias in the interpretation of our study findings. With our drug claims data, we are also unable to comment on the extent of unfilled prescriptions. Further, our measures of polypharmacy do not allow for clear conclusions regarding the appropriateness of pharmacotherapeutic care among our population of older women and men. Though beyond the scope of this study, additional research is also needed to examine sex-specific trends in particular clusters of diseases that may differentially impact changes in polypharmacy prevalence [22,30,32].

## Conclusions

Between 2003 and 2016 there was a significant increase in the prevalence of multimorbidity, polypharmacy and hyper-polypharmacy among adults aged 66 and older in Ontario, Canada. Notably, the impact of multimorbidity on changes in polypharmacy outcomes varied by age and sex. Whereas the prevalence of polypharmacy generally declined among women (especially younger women with fewer chronic conditions), it increased across all ages and multimorbidity levels among men. Though polypharmacy is not necessarily inappropriate, as the number of medications and regimen complexity increases among older adults with multiple chronic conditions, so does their potential risk for significant medication related harms [15,48,49]. Further investigations are needed to address potential sex and age differences in the clinical appropriateness of observed changes in polypharmacy levels. The differential changes we observed in polypharmacy trends in relation to multimorbidity level (i.e., relative increase in polypharmacy prevalence among men but a decrease among women) could reflect more or less appropriate prescribing in the management of chronic diseases or a mixture of both. Priority areas for future research include studies that expand our understanding of sex and gender differences in the drivers and health and societal outcomes of recent polypharmacy and multimorbidity trends in older populations.

## Supporting information

**S1 Table. Description of Ontario health administrative databases.**
(DOCX)

**S2 Table. Methods for ascertaining chronic conditions and level of multimorbidity.**
(DOCX)

**S3 Table. Characteristics of older adults (aged ≥66 years) in Ontario, Canada: 2003 vs 2016.**
(DOCX)

**S4 Table. Results (b, SE) from logistic regression analyses used to make predictions for polypharmacy outcome.**
(DOCX)

**S5 Table. Marginal probabilities of polypharmacy (5+ drugs) and unadjusted and adjusted risk differences and risk ratios, by select ages, level of multimorbidity and sex (excluding older adults residing in long term care facilities).**
(DOCX)

**S6 Table. Marginal probabilities of polypharmacy (5+ drug subclasses) and unadjusted and adjusted risk differences and risk ratios, by select ages, level of multimorbidity and sex.**
(DOCX)

**S7 Table. Most frequently dispensed drug subclasses for women and men (age ≥66 years) in Ontario, Canada in 2003 with polypharmacy and hyper-polypharmacy.**
(DOCX)

**S1 Text. RECORD statement.**
(DOCX)

## Acknowledgments

### Code availability

The full dataset creation plan and underlying analytic code are available from the authors upon request, understanding that the computer programs may rely upon coding templates or macros that are unique to ICES and are therefore either inaccessible or may require modification.

### Other acknowledgements

We thank IQVIA Solutions Canada Inc. for use of their Drug Information Database.

## Author Contributions

**Conceptualization:** Colleen J. Maxwell, Luke Mondor.

**Data curation:** Luke Mondor.

**Formal analysis:** Luke Mondor.

**Funding acquisition:** Walter P. Wodchis.

**Investigation:** Colleen J. Maxwell, Luke Mondor, Anna J. Pefoyo Koné, David B. Hogan, Walter P. Wodchis.

**Methodology:** Colleen J. Maxwell, Luke Mondor, Anna J. Pefoyo Koné, David B. Hogan, Walter P. Wodchis.

**Project administration:** Colleen J. Maxwell, Luke Mondor.

**Supervision:** Walter P. Wodchis.

**Writing – original draft:** Colleen J. Maxwell, Luke Mondor.

**Writing – review & editing:** Colleen J. Maxwell, Luke Mondor, Anna J. Pefoyo Koné, David B. Hogan, Walter P. Wodchis.

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
