## [Decision Letter · Decision Letter 0]

23 Feb 2021

PONE-D-21-02920

Sex differences in multimorbidity and polypharmacy trends: a repeated cross-sectional study of older adults in Ontario, Canada

PLOS ONE

Dear Dr. Maxwell,

Thank you for submitting your manuscript to PLOS ONE. After careful consideration, we feel that it has merit but does not fully meet PLOS ONE’s publication criteria as it currently stands. It is a well-written paper that covers an important topic. However, the reviewers highlight several aspects that need to be answered. Therefore, we invite you to submit a revised version of the manuscript that addresses the points raised during the review process.

We look forward to receiving your revised manuscript.

Kind regards,

Juan F. Orueta, MD, PhD

Academic Editor

PLOS ONE

Journal Requirements:

Reviewers' comments:

Reviewer's Responses to Questions

**Comments to the Author**

1. Is the manuscript technically sound, and do the data support the conclusions?

Reviewer #1: Yes

Reviewer #2: Yes

Reviewer #3: Partly

2. Has the statistical analysis been performed appropriately and rigorously? 

Reviewer #1: Yes

Reviewer #2: I Don't Know

Reviewer #3: Yes

3. Have the authors made all data underlying the findings in their manuscript fully available?

Reviewer #1: No

Reviewer #2: Yes

Reviewer #3: Yes

4. Is the manuscript presented in an intelligible fashion and written in standard English?

Reviewer #1: Yes

Reviewer #2: Yes

Reviewer #3: Yes

5. Review Comments to the Author

Reviewer #1: Review PONE-D-21-02920 - [EMID:81738d4af5e5d035]

Summary

The cross-sectional study examined the interrelationships between of age, gender, and multimorbidity with polypharmacy in the study participants. The study was conducted for two different study time points (2003 and 2016) in a canadian database to compare the change in polypharmacy over the years. This change was described in the study using various epidemiological measures (risk ratio, risk differnece etc.). Furthermore, multivariate logistic models analyzed the impact of various factors (monetary income, place of residence, etc.) on the increase in polypharmacy over time.

Major issues

1. The documents we received do not contain any information describing the study population or the distributions of the analyzed variables. The Table 1 mentioned in the text is unfortunately not included in the supplementary material. The absence of this information also means that points 13,14 and 15 of the STROBE statement are not fulfilled. Could the authors please add this information?

2. The authors have compared two different cohorts (all older than 65 years in 2003 were compared with all older than 65 years in 2016)? From our point of view, this warrants more detailed discussion how bias could have been introduced by comparing different cohorts from different points of time.

3. Could the authors please elaborate on their choice of chronic diseases as other chronic diseases, with many drugs used in therapy, such as HIV/AIDS or peripheral vascular occlusive disease, were not considered?

4. Why were some relevant ICD codes (e.g., for atrial fibrillation I48.2 or for osteoporosis M80) not included, although from our point of view they would be important for the correct identification of the diseases considered. Generally, could the authors comment on how the (validated) code sets were derived?

5. How did the authors deal with the fact that in the long period from 2003 to 2016, new drugs were steadily approved for the treatment of some of the classes they considered (in atrial fibrillation, DOAKs; in diabetes, dulaglutide, etc.)? If these newly approved drugs were not taken into account, this could distort the results, since new drug classes and combination options for the treatment of one and the same disease (and thus a large number of drugs) only exist in the late cohort.

6. Which drugs were considered in the study to determine polypharmacy? If all medications were considered on an ATC basis, this could distort the results, because also sporadically prescribed treatments such as antibiotics would have been used to determine polypharmacy.

Reviewer #2: Overall strength

• Overall well conducted and reported cross sectional analysis of a population based dataset looking at age and sex specific trends in both multimorbidity and polypharmacy.

Overall weakness

• It is difficult to infer the potential meaning of the result, as no explicit potentially inappropriate medication measures were applied to assess the quality of prescribing at both time points. The authors conclude that the development of specific male targeted deprescribing interventions may be an approach to address the rise in the prevalence of polypharmacy in men over the 13 year period. However, is it possible that this rising prevalence may represent improved prescribing and reduced potentially inappropriate omissions?

Specific comments

Methods

Line 137. Should the year be 2016 Vs 2003, (Not 2016 Vs 2013)?

Results

•Please include abbreviations at the end of tables.

• I feel more detail is needed on the age profile of the population cohort at both time points rather than just the median age with interquartile range, for example include the proportion and number of people in various age categories. Could some of the difference in 2016 be explained by changes in male life expectancy? A Danish prospective cohort reported (albeit over a shorter time period) reported an increase in the per person number of medicines in both sexes and that those who exited the study prematurely due to death had a higher number of medicines at entry and a greater increase in the number of medicines up to the time period before death (1).

• Would it be possible to include a similar figure to figure 1 showing the proportion with 0, 1, 2, 3 etc. medicines at the various age cuts. I think visualising the actual proportion for each number of medicines at both time points would help give the reader a sense of the magnitude of the change over the 13-year period?

Line 219, corticosteroids use seems very high. Can you specify if these were systemic or topical/inhaled?

Discussion

•The authors have clearly stated it is not possible to assess causation due to the cross sectional design but I feel possible reasons for these sex differences should be explored a little more particularly with respect to potentially inappropriate prescribing as this would add weight to the finding and might guide more specific recommendations for further research.

•I agree it would be interesting to correlate the findings with changes in PIP, which has generally declined when increases in polypharmacy are accounted for (2). Perhaps more men had potentially inappropriate omissions and more women had potentially inappropriate prescriptions (especially related to hypnotic and analgesia use). Perhaps these changes reflect improvements in prescribing over the 13-year study period.

References

1. Wastesson JW, Oksuzyan A, von Bornemann Hjelmborg J, Christensen K. Changes in Drug Use and Polypharmacy After the Age of 90: A Longitudinal Study of the Danish 1905 Cohort. Journal of the American Geriatrics Society. 2017;65(1):160-4.

2. Moriarty F, Hardy C, Bennett K, Smith SM, Fahey T. Trends and interaction of polypharmacy and potentially inappropriate prescribing in primary care over 15 years in Ireland: a repeated cross-sectional study. BMJ Open. 2015;5(9):e008656.

Reviewer #3: This manuscripts tackles the issues of prescription trends in elderly people from Ontario, comparing two periods, 2016 vs 2003. The authors reported a considerable use of

- In Backgrounds section, the main ideas are related with (i) an upward trend in multimorbidity, (ii) a correlation between coexistence of chronic diseases and medication balance (polypharmacy) and (iii) the lack of differences between sex groups in previous publications. However, too many references have been used and it is quite difficult to follow the flow of these ideas. Lines 66-68 can be answered with any of the systematic reviews driven by Prados-Torres (DOI:10.1016/j.jclinepi.2013.09.021) or C. Violan (https://doi.org/10.1371/journal.pone.0102149). In my opinion, last sentence of this section (line84-85) should be placed in Discussion section being developed more accurately.

- In Methods section: it should be defined the acronyms ICES (line 97)I could not find it in main document or supplementary files. Google helped me to know that is an Institution with health information from Ontario. I recommend to describe a brief description or add a reference because this info is added in “data availability” at the end of the document. Related with inclusion criteria, I was wondering if 65 years are included or people are older (66 years), consider using > or ≥, or +65 years as it was used to describe multimorbidity and medication use (homogenized).

-In Results sections I suggest some points.

oIt can be considered a row in Table 1 with relative differences between 2016 and 2003 to evaluate trends between both time points.

oI suggest in Line 192 and 198 homogenize “those with no or 1 condition” by “0/1 condition”

o Results of polypharmacy and hyperpolypharmacy from 2003 (s5) is quite confusing. It is said that NSAIDs and BDZ are declined, I guess they are “declined compared to hyperpolypharmacy group”. May be confusing the meaning of this findings. What about other differences like corticosteroids or diuretics? They seemed to have the same trend and they are not mention.

- In Discussion section, the text brings all the information into a coherent whole, connecting the findings with the aim of the study. However, authors interpretation is missing at some point. For example, why NSAIDs and BDZ are common in older people with hyperpolypharmacy rather than polypharmacy, they are included in STOP/START criteria and they are still very common. Besides, scarcely clinical meaning is discussed. Women have a longer expectancy of living, they have different multimorbidity patters compared with men because of gender-related diseases and lifestyle (for instance, smoking or alcohol habit (DOI: 10.1016/j.jclinepi.2013.09.021). In my opinion, discussion results are quite in the surface and could be improved.

- On the other hand, it has been interesting to work with most common chronic conditions because it is easier to understand the relation of data. However, heterogeneity in chronic condition combination is not represented in this article. In limitation section it should be considered the few number of chronic conditions considered. Multimorbidity is probably higher because of this limitation (DOI: 10.1016/j.jclinepi.2013.09.021).

- Finally, the conclusion should be more specific. It might end with a strong statement but is quite redundant with results found. It contains references and in my opinion, references should be placed in discussion section. This section need to emphasize contribution of your work and may be it implications for future research.

6. PLOS authors have the option to publish the peer review history of their article (what does this mean?). If published, this will include your full peer review and any attached files.

Reviewer #1: No

Reviewer #2: No

Reviewer #3: **Yes: **Marina Guisado-Clavero

---

## [Author Response · Author response to Decision Letter 0]

24 Mar 2021

Response to Reviewer File has been uploaded.

---

## [Decision Letter · Decision Letter 1]

12 Apr 2021

Sex differences in multimorbidity and polypharmacy trends: a repeated cross-sectional study of older adults in Ontario, Canada

PONE-D-21-02920R1

Dear Dr. Maxwell,

We’re pleased to inform you that your manuscript has been judged scientifically suitable for publication and will be formally accepted for publication once it meets all outstanding technical requirements.

Kind regards,

Juan F. Orueta, MD, PhD

Academic Editor

PLOS ONE

Reviewers' comments:

Reviewer's Responses to Questions

**Comments to the Author**

1. If the authors have adequately addressed your comments raised in a previous round of review and you feel that this manuscript is now acceptable for publication, you may indicate that here to bypass the “Comments to the Author” section, enter your conflict of interest statement in the “Confidential to Editor” section, and submit your "Accept" recommendation.

Reviewer #1: All comments have been addressed

Reviewer #2: All comments have been addressed

Reviewer #3: All comments have been addressed

2. Is the manuscript technically sound, and do the data support the conclusions?

Reviewer #1: Yes

Reviewer #2: Yes

Reviewer #3: Yes

3. Has the statistical analysis been performed appropriately and rigorously? 

Reviewer #1: Yes

Reviewer #2: I Don't Know

Reviewer #3: Yes

4. Have the authors made all data underlying the findings in their manuscript fully available?

Reviewer #1: No

Reviewer #2: Yes

Reviewer #3: Yes

5. Is the manuscript presented in an intelligible fashion and written in standard English?

Reviewer #1: Yes

Reviewer #2: Yes

Reviewer #3: Yes

6. Review Comments to the Author

Reviewer #1: Comments have been sufficiently addressed.

I have no concerns about research ethics or publication ethics; however, it's Always a pity when data cannot be made available (for which the authors are not responsible, though).

Reviewer #2: Thank you for addressing and replying to my my comments. The authors have improved the manuscript by further commenting on the potential causes for these sex differences and specific areas that warrant further research.

Reviewer #3: Dear Corresponding author and authorship,

I would congratulate the effort and responses given to all reviewers. From my point of view, my suggestions have been addressed and have been considered or refused properly. I hope in future research we could answer those reminding gaps. I checked the all manuscript with track change and I did not have more suggestions, may be consider removing the hyperlink characteristics (blue color) from those web sides attached in Reference section.

Kind regards,

7. PLOS authors have the option to publish the peer review history of their article (what does this mean?). If published, this will include your full peer review and any attached files.

Reviewer #1: No

Reviewer #2: **Yes: **Caroline McCarthy

Reviewer #3: No

---

## [Editor Report · Acceptance letter]

15 Apr 2021

PONE-D-21-02920R1 

Sex differences in multimorbidity and polypharmacy trends: a repeated cross-sectional study of older adults in Ontario, Canada 

Dear Dr. Maxwell:

I'm pleased to inform you that your manuscript has been deemed suitable for publication in PLOS ONE. Congratulations! Your manuscript is now with our production department. 

Kind regards, 

on behalf of

Dr. Juan F. Orueta 

Academic Editor

PLOS ONE